# Nutritional, Phytochemical, and Antimicrobial Properties of *Carica papaya* Leaves: Implications for Health Benefits and Food Applications

**DOI:** 10.3390/foods14020154

**Published:** 2025-01-07

**Authors:** Rajni Choudhary, Ravinder Kaushik, Ansab Akhtar, Suvendu Manna, Jyoti Sharma, Aarti Bains

**Affiliations:** 1School of Health Sciences and Technology, University of Petroleum and Energy Studies (UPES), Dehradun 248007, Uttarakhand, India; rajnichoudhary9372@gmail.com; 2School of Medicine, Louisiana State University, New Orleans, LA 70112, USA; ansabakhtar@gmail.com; 3Sustainability Cluster, School of Advance Engineering, University of Petroleum and Energy Studies (UPES), Dehradun 248007, Uttarakhand, India; smanna@ddn.upes.ac.in; 4Department of Botany, Baba Mastnath University, Rohtak 124001, Haryana, India; jyotijiten09@gmail.com; 5Department of Microbiology, Lovely Professional University, Phagwara 144411, Punjab, India; aarti05888@gmail.com

**Keywords:** papaya leaves, phenolic compounds, phytochemicals, antioxidant activity

## Abstract

Background: Papaya leaves (PLs) are known for their therapeutic benefits and traditional use in treating inflammation, infections, and various health conditions. Rich in bioactive compounds, PLs are studied for their potential applications in functional foods. This study analyzed their nutritional, phytochemical, structural, thermal, and antimicrobial properties to evaluate their role as a health-promoting ingredient. Methods: Phytochemicals were quantified spectrophotometrically and identified via GC-MS. Antioxidant activity was assessed using DPPH and FRAP assays. Mineral content was determined using ICP-OES. Structural and thermal properties were evaluated using FTIR, XRD, and calorimetry, and antimicrobial activity was tested via the agar well diffusion method. Results: PLs contained 25.75% crude protein, 41.49% carbohydrates, and high levels of flavonoids (21.00 mg QE/g), phenolics (8.85 mg GAE/g), and tannins (430 mg TAE/g). Antioxidant assays confirmed strong free radical scavenging potential. Mineral analysis showed abundant K, Ca, Mg, Na, and Fe (4071, 1079, 789.2, 361.2, and 228.2 mg/kg, respectively). Structural and thermal analysis revealed bioactive functional groups, 23.9% crystallinity, and thermal degradation characteristics. PLs exhibited antimicrobial activity, inhibiting *E. coli*, *S. aureus*, *B. subtilis*, and *K. pneumoniae* with zones of 22.05–25.15 mm. Conclusions: PLs demonstrate strong nutritional, antioxidant, and antimicrobial properties, supporting their inclusion in functional food.

## 1. Introduction

The *Carica papaya* plant mostly grows in the tropical and subtropical regions of Australia, Brazil, China, India, Malaysia, Myanmar, and the Philippines. This plant belongs to the *Caricacea* family and is famous for its fruit; however, the leaves of the plant are also a treasure of health. Traditionally, papaya leaves (PLs) have been used worldwide in medicinal formulations to treat several diseases like dengue, malarial infection, cancer, and diabetes, owing to the existence of bioactive compounds such as catechins, flavonoids, tocopherol, alkaloids, anthocyanins, tannins, saponins, and phenols [1,2]. The use of PLs as a traditional remedy is due to their anti-cancer, antiseptic, hypoglycemic, antiviral, and wound-healing properties [3]. PLs are an abundant source of minerals, including calcium (Ca), potassium (K), magnesium (Mg), zinc (Zn), manganese (Mn), and iron (Fe) [4]. Their higher iron content makes them a useful source to treat anemia, tuberculosis, and growth disorders and act as a cleanser in herbal drugs [5,6]. PLs exhibit excellent antioxidant properties due to their secondary metabolites, such as flavonoids and phenols [7]. Being abundant in fiber, ascorbic acid, and antioxidants, PLs prevent the deposition of cholesterol in the arteries, which may lead to arthritis, and can reduce the risk of cardiovascular diseases, aging, macular degradation, and cancer. In addition, they play a critical role in the functioning of the nervous system and heart, blood coagulation, and muscle movement [8]. PLs show antibacterial properties against several Gram-positive and Gram-negative bacterial species such as *Bacillus cereus*, *Klebsiella pneumonia*, *Micrococcus luteus*, *Escherichia coli*, *Pseudomonas aeruginosa*, and *Staphylococcus aureus* by inhibiting bacterial protein synthesis, which further highlights their potential in diverse applications [9].

Despite these established medicinal uses, the application of PLs in functional food formulations remains an emerging area of interest. Plant-based products, including PLs, offer several advantages over synthetic alternatives, such as affordability, fewer side effects, and wide availability. While previous studies have explored the nutritional and medicinal properties of PLs, this study aims to contribute new insights by investigating their thermal properties, crystallinity, and bioactive compounds using advanced analytical techniques like DSC, XRD, FTIR, and GC-MS. These analyses provide new data on the behavior of PLs under various conditions, which is crucial for their potential application in the food industry. This practical approach not only highlights their value in functional food formulations but also offers a fresh perspective on their application beyond traditional medicinal uses. Additionally, the antibacterial effects of PLs were evaluated against both Gram-negative and Gram-positive bacterial strains, providing a further rationale for their inclusion in food formulations and other food-related applications.

## 2. Materials and Methods

### 2.1. Materials

Fresh young PLs at the tender stage were collected from the local village of Dehradun (Bidholi), Uttarakhand, India in January 2024. The PL sample underwent cleaning with distilled water to remove dust particles, followed by tray drying (DM, Dellmarc, Kerala, India) at a temperature of 55 °C for 6 h. The dried PLs were ground into a fine powder using a mechanical grinder (Sujata, Delhi, India), and were then sieved through a 60–80 mesh sieve to obtain a uniformly sized sample. The prepared powder was stored in an airtight vessel and refrigerated at 4 °C for 24 h to maintain freshness before further analysis. For the experimental procedures, nitric acid (AR, 99%), sodium carbonate (AR, 99.5%), potassium acetate (ACS, 99.0%), hydrogen peroxide (AR, 30% *w*/*v*), sodium hydrogen carbonate (ACS, 99.0%), phenol (AR, ≥99.0%), petroleum ether (HPLC grade, 99%), bromocresol green (LR, 98%), copper sulfate (LR, 99%), boric acid (LR, 99.9%), sulfuric acid (AR, 98%), methanol (AR, ≥ 99.8%), hydrochloric acid (AR, 99%) were procured from Rankem, Avantor Performance Materials India Ltd., Mumbai, Maharashtra India. Gallic acid (ACS, 98%), 2, 2 diphenyl-1-picrylhydrazyl (95%), Folin–Ciocalteu phenol reagent (AR, 2.0N), bovine serum albumin (98%), sodium potassium tartrate (AR, 99%), potassium bromide (AR, 99.5%), ferric chloride (98%), and aluminum chloride (AR, 97%) were acquired from SRL, Mumbai, India. Additionally, quercetin (HPLC, 95%), 2, 4, 6-Tris (2-pyridyl)-s-triazine (≥98%), acetic acid (ACS, 100%), tannic acid (ACS), and ammonium hydroxide (ACS, 30%) were obtained from Merk Life Science PVT. Ltd., Mumbai, India. For the microbial study, Mueller–Hinton agar from Hi-Media, Mumbai, India, and the bacterial strains *E. coli* (MTCC 2126), *S. aureus* (MTCC 96), *B. subtilis* (MTCC 441), and *K. pneumoniae* (MTCC 2403) were obtained from the Institute of Microbial Technology (IMTECH), Chandigarh, India. To maintain the highest level of quality, all chemicals employed in this analysis were of analytical grade and did not require further purification.

### 2.2. Methods

#### 2.2.1. Sample Extraction

For total phenols, flavonoids, antioxidant activity, GC-MS, and antimicrobial analysis, extraction was performed by soaking 1 g of dried papaya leaf powder in 10 mL of 70% methanol (1:10 *w*/*v*) for 8 h at 25 °C using an orbital shaker (REMI, CIS-24 PLUS, Mumbai, India) set at 200 rpm. Subsequently, the extract was filtered using Whatman No. 40 filter paper. The obtained filtrate was concentrated under reduced pressure in a rotary evaporator at 45 °C to remove excess solvent and obtain a more concentrated extract for further analysis. The extract was stored in amber bottles at 4 °C for further use. Meanwhile, for the analysis of total soluble protein, distilled water was used as the solvent.

#### 2.2.2. Nutritional Composition

##### Proximate Analysis

The PL powder was subjected to moisture (oven drying), fat (soxhlet method), crude protein (Kjeldahl method), total soluble protein (TSP) (Lowry method), carbohydrates (phenol-sulfuric acid method), ash, and crude fiber content analyses, as per the methods described in AOAC, 2005 [10].

##### Mineral Analysis

Inductively coupled plasma optical emission spectroscopy (ICP-OES) was used to estimate the Ca, Cr, Cu, Fe, K, Mg, Mn, Na, and Zn contents in PLs, following the method outlined by Kiani et al. [11]. About 2 g of PLs was incinerated to ash in a muffle furnace at 550 °C for 4 h. Then, 9 mL of HNO_3_ and 1 mL of H_2_O_2_ (9:1 *v*/*v*) were added to the ash and held for 10 min at 25 °C to homogenize the sample. After acid digestion, the sample was heated on a heating mantle inside a fume hood for 15 min. The resulting solution was evaporated to eliminate excess acid until a semi-dried mass was obtained, after which the final volume was adjusted to 20 mL by adding deionized water in a volumetric flask.

#### 2.2.3. Phytochemical Screening

##### Total Phenolic Content (TPC)

TPC was evaluated using the procedure developed by Aryal et al. [12]. Briefly, the sample extracts were mixed with 95% (1 mL) methanol, 10% Folin–Ciocalteu reagent (5 mL), and 7% sodium carbonate (4 mL). The reaction mixture was placed in a water bath and incubated at 30–35 °C for 30 min. The absorbance was read at 760 nm using a UV-visible spectrophotometer (LAMDA 35, PerkinElmer Inc., Waltham, MA, USA) against a blank. Gallic acid was used to construct the standard curve, and total phenolic content was expressed as the gallic acid equivalent (mg GAE/g).

##### Total Flavonoid Content (TFC)

TFC was evaluated by employing an aluminum chloride (AlCl_3_) colorimetric assay, as reported by Zin et al. [13]. First, 10% AlCl_3_ (0.1 mL), 1 M potassium acetate (0.1 mL), and 95% methanol (1.5 mL) were mixed with 0.5 mL of the sample. The reaction mixture was left at 25 °C for 30 min, after which 2.8 mL of distilled water was added. The absorbance was read at 415 nm using a UV-visible spectrophotometer. Quercetin was used as a reference standard to prepare a standard curve, and the TFC was indicated as the mg quercetin equivalent (mg QE/g).

##### Tannin Content

The tannin content of the PLs was estimated by employing the procedure outlined by Kumar et al. [14]. Briefly, 250 mg of the sample was heated in 20 mL of distilled water for 30 min at 25 °C and then filtered through Whatman No. 40 filter paper. The resulting filtrate was mixed with 0.25 mL of Folin–Ciocalteu reagent, 1.25 mL of Na_2_CO_3,_ and 0.50 mL of distilled water. The reaction mixture was incubated for 40 min at 25 °C, followed by measurement of the absorbance at 725 nm using a UV-visible spectrophotometer. A standard curve was prepared, using tannic acid as a reference standard.

##### Alkaloid Content

The alkaloid content in the PLs was estimated using the method followed by Mir et al. [15], with slight modifications. A solution of 10% acetic acid in ethanol (200 mL) was mixed with 5 g of the sample, covered, and left to stand for 4 h at 25 °C. The mixture was subsequently filtered through Whatman No. 40 filter paper and concentrated to one-quarter of its initial volume using a water bath at a temperature of 50–60 °C. Concentrated ammonium hydroxide (NH_4_OH) was added dropwise to the concentrated filtrate until precipitation was finished. The precipitate was collected, washed with dilute NH_4_OH, and filtered, and the resulting alkaloid residue was dried and weighed. The results were expressed in g/100 g of the dry weight of the sample.

##### GC-MS Analysis

The methanolic PL extract was subjected to GC-MS analysis using a Clarus 500 Perkin Elmer system, coupled with a Turbo mass gold mass detector (PerkinElmer Inc.,Waltham, MA, USA) and equipped with an Elite-1 column PE5 (100% dimethylpolysiloxane, 30 m × 0.25 mm × 0.25 µm). Initially, the instrument temperature was maintained at 110 °C for 2 min. The temperature was then raised at a rate of 5 °C/min until it reached 280 °C, which was held for 9 min. The injection port temperature was kept at 250 °C, with a helium flow rate of 1 mL/min. The ionization voltage was set to 70.00 eV. Samples were injected in split mode 1, and the mass spectrometer scan range was set from 40 to 450 amu. The mass spectra fragmentation patterns were matched against entries in the NIST (National Institute of Standards and Technology) library database for comparison. The amount of each component was determined based on its relative peak area in the chromatogram.

#### 2.2.4. Assessment of Antioxidant Activity

##### DPPH Assay

The PL extract was assessed for its free radical scavenging activity using a 2,2 diphenyl-1-picrylhydrazyl radical (DPPH) assay, as reported by Flieger et al. [16]. A volume of 1 mL of the sample was mixed with 80% methanol (4 mL), and 1 mL (containing 1 mmole) of DPPH. The reaction mixture was incubated for 20 min at 25 °C, and the OD (optical density) was measured at 517 nm using a UV-Vis spectrophotometer, to calculate the µmoles of DPPH scavenged by the leaf extract. The radical scavenging activity was determined using the formula below and reported as a percentage.
$$\%\;DPPH\;radical\;scavenging\;activity=\frac{\left(OD\;control-OD\;sample\right)}{OD\;control}\times 100$$

##### FRAP Assay

The ferric-reducing anti-oxidant power (FRAP) assay was conducted, following the method reported by Nisa et al. [17] with slight modifications. Concisely, the FRAP reagent was prepared using 0.1 M acetate buffer (75 mL), 10 mM 2,4,6-tripyridyl-S-triazine (TPTZ) (7.5 mL) (prepared in 40 Mm HCl), and 20 Mm ferric chloride (7.5 mL). Approximately 0.5 mL of the sample extract was mixed with 3.5 mL of the FRAP reagent and then incubated in a water bath at 37 °C for 30 min. The absorbance was read at 593 nm using a UV-Vis spectrophotometer, and the results were calculated as equivalents to Fe^2+^ using a standard curve equation based on FeSO_4.7_H_2_O, with concentrations ranging from 4 to 20 mol/mL.

#### 2.2.5. FTIR Analysis

The dried PL powder underwent FTIR analysis, following the procedure outlined by Sutariya et al. [18]. In this procedure, 100 mg of potassium bromide (KBr) is mixed with 10 mg of leaf powder to prepare a translucent sample disc under a hydraulic pallet press. The prepared pellet was loaded into an FTIR spectroscope (Perkin Elmer, Frontier FT-IR/FIR) with a spectral range of 4000–500 cm^−1^ and a resolution of 0.4 cm^−1^.

#### 2.2.6. XRD Analysis

The Bruker D8 Advance Eco diffractometer (Vertical, Theta/Theta, or Theta/2Theta geometry) was employed for this study. The tightly packed fine sample was fixed into an X-ray diffractometer and copper Kα, 2λ (λ = 1.540 µm −1.544 Å:40 kv: 25 mA), was set to generate the X-ray pattern. The scan was generated from 5 to 80° at a step size of 0.02 and a count time of 3 s. The percentage of crystallinity was calculated using the Gaussian fit method, while the crystalline size was determined using the Debye–Scherrer equation.
$$D=\frac{\kappa \lambda }{\beta cos\theta }$$

The crystalline size was represented by *D*, which is likely to be equal to or shorter than the grain size. *K* represents the shape factor, *λ* stands for the x-ray wavelength, *β* indicates the line broadening at half the maximum intensity in radians, and *θ* is the Bragg angle [19].

#### 2.2.7. Thermal Properties

The thermal properties and phase change behavior of PL powder were assessed using a differential scanning calorimeter (DSC-7020, Hitachi High-Tech Science Corporation, Tokyo, Japan). About 5 mg of the sample was placed in an aluminum pan and scanned at temperatures ranging from 25 to 300 °C, with a heating rate of 10 °C/min, under a nitrogen atmosphere to prevent the oxidation of the sample. Onset, end-set, melting peak temperature, and melting enthalpy were obtained accordingly, using NEXTA software STA 200.

#### 2.2.8. Antimicrobial Activity

The methanolic extract of PLs was tested for antimicrobial activity using the well diffusion method with slight modifications, as described by Joshi et al. [20]. Antimicrobial activity was assessed against the Gram-positive *S. aureus*, and *B. subtilis*, as well as the Gram-negative *E. coli* and *K. pneumoniae* bacterial strains, by observing the zone of inhibition. Bacterial inoculums were swabbed across the surface of the nutrient agar medium using a sterile cotton swab to ensure consistent bacterial growth on the Petri plate. About 100 µL of the PL extract was loaded into the wells and incubated for 24 h at 37 °C. Subsequently, the inhibition zone that formed on the Petri plate was measured.

#### 2.2.9. Statistical Analysis

The experiments were carried out in triplicate, and the results were indicated as the mean ± standard deviation of the observations. Statistical analysis was conducted using an ANOVA in Microsoft^®^ Excel, 2016 (Microsoft Corporation, Redmond, WA, USA), with significance set at *p* < 0.05. Graphical analysis was carried out using OriginPro 8.5 (OriginLab Co., Northampton, MA, USA).

## 3. Results and Discussion

### 3.1. Nutritional, Phytochemical, and Antioxidant Activity

The nutritional and phytochemical composition of PLs is presented in Table 1. The moisture content in PLs was 15.16 g/100 g, and the ash content was 3.73 g/100 g; in contrast, previous studies reported ash contents in PLs ranging from 2.25 to 2.18% [21,22]. The fat content was 4.50 g/100 g. The crude fiber content was 9.06 g/100 g, although Olumide et al. [5] reported a range of 5.70 to 8.95% for crude fiber content in dry and fresh PLs, respectively. The crude protein and carbohydrate content of the PLs was 25.75 g/100 g and 41.49 g/100 g, respectively. However, Joseph et al. [23] observed a higher value for protein (29.5%). In contrast, the carbohydrate content observed in this study was lower than the 58.3% reported by Dev et al. [24]. The TSP content was 52.37 mg/g, which highlights the potential of PLs as a source of bioavailable protein. This protein contributes to numerous digestive and therapeutic benefits and is absorbed and utilized directly by the body. The results revealed that crude protein and carbohydrates are the major nutritional components of PLs. The protein in PLs could be used to supplement other sources of protein. PLs containing crude fiber are beneficial for regular bowel movement and aid in nutrient absorption. According to some research, the consumption of PL fiber helps balance blood sugar and lower cholesterol levels [25]. The mineral content evaluation revealed that the Ca content in PLs was 1079 mg/kg. According to the DRIs (dietary reference intakes), the recommended dietary allowance (RDA) value of Ca for males and females is reported to lie between 1000 and 1300 mg/day [26]; therefore, it can be utilized for calcium-enriched food product formulations. Similarly, Cu (13.24 mg/kg), Fe (228.2 mg/kg), K (4071 mg/kg), Mg (789.2 mg/kg), Zn (70.92 mg/kg), and Na (361.2 mg/kg) were observed. The Mn content was (22.15 mg/kg), whereas Sharma et al. [27] described 22.88 mg/kg Mn in PLs. Among all the minerals, K was the predominant element. A high concentration of K is believed to enhance Fe utilization, and, in addition, to regulate blood pressure through body fluid regulation [28]. Furthermore, Mg assimilates phosphorous, and Na participates in the metabolism of water, cleanses the digestive tract, facilitates osmosis and assimilation, promotes digestion, and combats stomach acidity, as well as alkalizing the blood [29]. Mn takes part in brain function and pineal gland and pituitary gland function, and also promotes hepato-renal functioning [30]. For this reason, PLs can be used for manufacturing mineral supplements, food enrichment, and fortification.

The TPC recorded a mean content of 8.85 mg GAE/g and the flavonoid content was 21.00 mg QE/g. Another study by Palanisamy and Basalingappa [31] reported a flavonoid content of 21.06 mg QE/g in PLs. The tannin content and alkaloid content in the PLs were 430 mg TAE/g, and 11.4 g/100 g, respectively. In contrast, Ugo et al. [22] reported 310 mg/100 g of tannin content in methanolic PL extract.

Furthermore, Figure 1 shows the GC-MS chromatogram of the methanolic extract of the PLs, displaying the retention time (RT), height, and area of each peak. Approximately 17 phytochemicals were identified through the GC-MS analysis. The names, RT, biological effects, the relative abundance that was indicated in terms of peak area (%), and multiple detections of the same compound at different RTs are presented in Table 2. The major phytochemicals in the methanolic extract of PLs were hydroperoxide, hexyl (15.69%), 9,12,15-octadecatrienal (11.25%), cyclohexasiloxane, decamethyl (5.51%), heptane,4-azido (3.48%), cyclopentasiloxane, and decamethyl (3.32%). Additionally, methane, isothiocyanato (1.41%), 1,1,1,3,5,5,7,7,7-nonamethyl 3 (trimethyl siloxy) tetrasiloxane (2.90%), 4-aminosalicylic acid, 3TMS derivative (2.69%), 1,1,1,5,7,7,7-heptamethyl-3,3 bis (trimethylsiloxy) tetrasiloxane (2.05%), piperazine, 1-nitroso (1.21%), 1,1,1,3,5,7,7,7-octamethyl-3,5-bis (trimethylsiloxy) tetrasiloxane (1.09%), ethanethioic acid, S-(2-methyl butyl) ester (2.99%), hexasiloxane, tetradecamethyl (1.11%), and 1H-1,2,4-triazole-3-carboxaldehyde 5-methyl (1.07%) were also present. These compounds are largely responsible for the antioxidant, antibacterial, and antifungal properties of PLs.

The antioxidant activity was assessed based on the radical scavenging capacity of DPPH and FRAP. The observed antioxidant activity was 77.55% and 25.34 mmol/mg for the DPPH and FRAP assays, respectively (Table 1). In contrast, previous studies reported 77.40% of radical scavenging activity of PLs by the DPPH assay, and 24.65 mmol/mg by the FRAP assay [17]. The GC-MS analysis revealed that the hydroperoxide, hexyl, and 9,12,15-octadecatrienal were major constituents in the methanolic extract of PLs, which likely contributes to their antioxidant activity. These compounds play a key role in reducing the oxidation of other molecules, thereby protecting the human body against damage from ROS (reactive oxygen species).

### 3.2. FTIR Findings

Figure 2 represents the FTIR spectrum of PL powder, showing a sharp and strong peak at 3433 cm^−1^, which can be attributed to the O-H stretching vibration commonly associated with hydroxyl groups in polyphenols and fibers, indicating the presence of polyphenolic compounds and fiber. Polyphenols are well known for their antioxidant, anti-inflammatory, and antibacterial properties, which are crucial for the therapeutic potential of PLs, while the peaks at 2925 cm^−1^, and 2848 cm^−1^ indicate C-H symmetric and asymmetric stretching vibrations in aliphatic chains. The fatty acids usually contain these methyl groups [47,48,49,50]. The peak at 1639 cm^−1^ is assigned to the amide I band of the protein group, which is equivalent to the C=O carbonyl asymmetric stretch and indicates the presence of proteins in the PLs, which are crucial for the nutritional value of the leaves, providing essential amino acids for human consumption. The observation of the amide III bands at 1409 cm^−1^ further supports the presence of proteins and reveals information about their secondary structure, such as the α-helix and β-sheet conformation. This is particularly useful for understanding the functional properties of proteins, including their potential for bioactive functions like enzyme inhibition or antioxidant activity [50,51]. The peak at 1029 cm^−1^ can be attributed to the C-O-C bond, indicating the presence of ether, carboxylic acid, and esters. These suggest a broad variety of bioactive compounds, including tannins, phenols, and flavonoids [52,53,54]. These compounds are well known for their antioxidant, antimicrobial, and antidiabetic properties, further highlighting the therapeutic potential of PLs. The presence of these bioactive compounds supports the claim that PLs could be used in the food and pharmaceutical industries for their functional benefits.

### 3.3. XRD Data and Observation

An X-ray diffraction study was employed to examine the crystalline properties of PL powder. The existence of large and diffuse peaks illustrates the amorphous nature of the materials in X-ray diffraction [55]. A prominent 2θ peak at 21.75 (Figure 3A) was assigned to the cellulose components [56,57], and the average crystalline size and degree of crystallinity were determined to be 1.16 nm and 23.9% respectively. This suggests that while the PLs have an overall amorphous structure, they still contain some crystalline cellulose, a major structural polysaccharide found in plant cell walls, which contributes to the mechanical strength and rigidity of the leaves. This relatively low degree of crystallinity is indicative of a highly amorphous structure in the PLs. Similar observations of *Moringa oleifera* and *Ananas comosus* leaf powders are described in the studies conducted by Sakr et al. [58] and Nadirah et al. [59]. These results indicate that the PLs have a naturally amorphous structure. This type of amorphous material can be used in the food industries to provide a smooth texture, enhance the solubility and bioavailability of nutrients, and improve sensory attributes due to better retention of flavor and aroma compounds [60,61,62]. For instance, nutrients and bioactive compounds in PLs, such as polyphenols and vitamins, may be more easily absorbed by the body, due to the amorphous matrix facilitating better solubility. Additionally, they offer rapid dissolution and can serve as effective carriers of bioactive compounds that are sensitive to light, heat, and oxygen, thereby enhancing the shelf-life of products.

### 3.4. Thermal Analysis

The thermal analysis results, obtained through DSC, provide significant insights into the thermal stability, composition, and processing characteristics of PL powder, which are directly related to its nutritional and phytochemical properties (Figure 3B). The initial endothermic peak and its melting peaks appeared at 119 °C with an onset at 97.75 °C, an end-set of 115.88 °C, and an enthalpy (ΔH) of 369.96 J/g, indicating the loss of volatile components such as water from the sample [2]. This indicates the moisture content in PLs and highlights the temperature range at which water is released, which is crucial for understanding how moisture impacts the nutritional content and bioavailability of the leaf’s compounds. A higher value of enthalpy indicates that greater energy was expended in the breakdown of non-covalent bonds. The second and third peaks were observed at approximately 147.91 °C (onset 147.66 °C and end-set 148.27 °C), and 177 °C (onset 185.63 °C and end-set 196.30 °C), with enthalpies ΔH of 42.46 J/g, and 25.18 J/g, respectively. These endothermic occurrences can be attributed to the melting of the functional constituents, generally polyphenols, within a temperature range of 130 to 270 °C [63]. These results emphasize the thermal stability of polyphenols, revealing their resilience at higher temperatures and supporting their potential use in health-related applications. The fourth exothermic peak was observed at 333.91 °C with an onset of 331.51 °C, an end-set of 335 °C, and an enthalpy of ΔH of 22.18 J/g, which is related to the degradation of the cellulose and its carboxyl groups [64]. In contrast, Huang et al. [65] reported cellulose degradation enthalpy ΔH 23.0 J/g in raw corn stalks. The relatively low enthalpy value of 22.18 J/g for cellulose degradation suggests that PLs are more thermally sensitive compared to other plant materials, such as corn stalks, which require higher energy for cellulose breakdown. This may influence the structural integrity of PLs under heat, making them more susceptible to degradation during processing, particularly in applications that involve heat treatment. These findings are linked to the crude fiber content in PLs, as cellulose is a major fiber component, and they help explain the behavior of the fiber during thermal treatments. Overall, the thermal analysis helps us understand the thermal processing parameters that preserve the nutritional and functional properties of PLs, ensuring their potential use in food processing and phytochemical applications while retaining their bioactive compounds.

### 3.5. Antimicrobial Activity of PLs

In this study, a methanolic PL extract was tested for its antimicrobial activity against *E. coli*, *S. aureus*, *B. cereus*, and *K. pneumoniae*, as illustrated in Figure 4. The PL extract showed equivalent antibacterial activity against all bacterial strains compared to the respective standard, ciprofloxacin. As a positive control, ciprofloxacin showed a larger inhibition zone of 27.38 ± 0.31, while methanolic PL extract showed a significantly (*p* < 0.05) higher inhibition zone against the Gram-positive *S. aureus* (25.15 mm) and *B. subtilis* (25.00 mm) than that with Gram-negative bacteria. Zones of inhibition with diameters of 23.18 mm and 22.05 mm were observed for *E. coli* and *K. pneumoniae*, respectively. These findings align with those recorded in Ref. [66], where a methanolic PL extract at a concentration of 100 µL showed a 20 mm zone of inhibition for *E. coli* and 24 mm for *S. aureus.* Similarly, Khadam et al. [67] reported a 24.6 mm inhibition zone for *B. subtilis* and 26.6 mm for *K. pneumoniae.* Several phytochemicals, as reported in the GC-MS analysis, such as methane isothiocyanato-, cyclopentasiloxane, decamethyl, ethanethioic acid, S-(2-methyl butyl) ester, and heptane, 4-azido- are attributable for the antimicrobial activity of PL extract. These compounds are known to cause the leakage of cytoplasmic constituents and inhibit bacterial cell wall formation [68]. In addition, phenolic compounds can be complex, with extracellular materials such as proteins and the cell walls of bacteria. Therefore, these results suggest that methanolic PL extract could be utilized in the production of antimicrobial food packaging, preservatives, and biodegradable films/coatings.

## 4. Conclusions

This study highlights the rich nutritional and phytochemical profile of PLs, including carbohydrates, proteins, fibers, minerals, and bioactive compounds with strong antioxidant properties. FTIR analysis confirms the presence of functional groups associated with bioactive compounds, while XRD reveals their amorphous nature, making PLs suitable for food applications such as flavor enhancement, solubility improvement, texture modification, and nutrient fortification. Thermal analysis shows bioactive compound degradation between 147 and 177 °C. The methanolic PL extract exhibits antimicrobial efficacy against both Gram-negative and Gram-positive microorganisms, which can be attributed to its phytochemicals. These findings highlight the potential of PLs as nutraceutical ingredients in food, although their functionality and consumer acceptability must be carefully considered.

## Figures and Tables

**Figure 1 foods-14-00154-f001:**
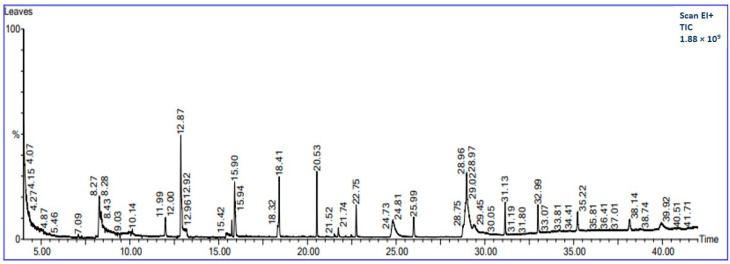
GC−MS chromatogram of methanolic papaya leaf extract.

**Figure 2 foods-14-00154-f002:**
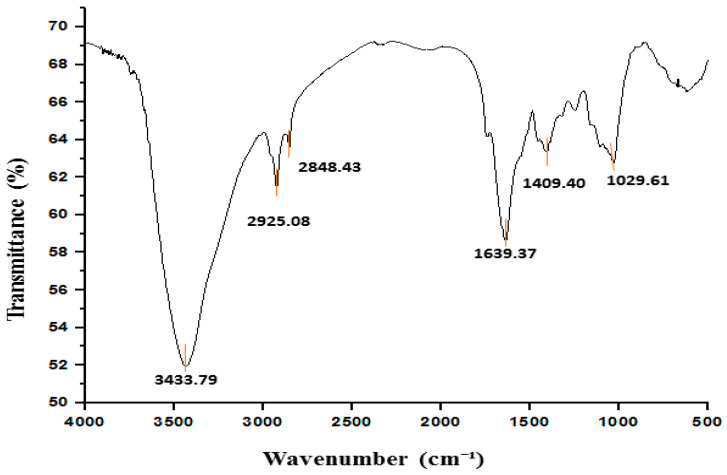
FTIR spectra of papaya leaf powder.

**Figure 3 foods-14-00154-f003:**
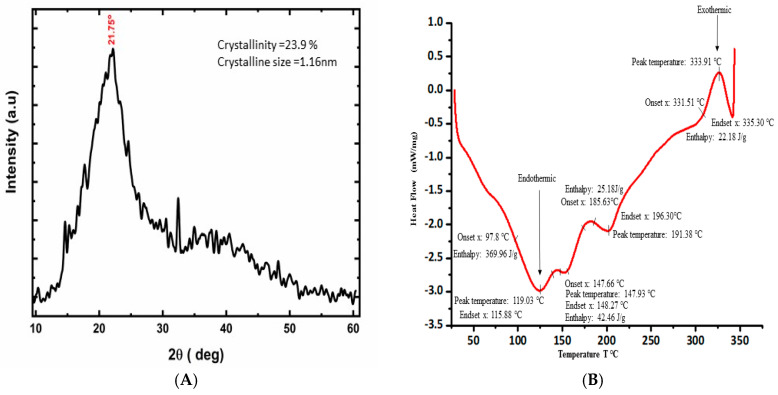
(**A**) X−ray diffraction pattern of papaya leaf powder. (**B**) DSC profile of papaya leaf powder in the temperature range of 25−300 °C, with a heating rate of 10 °C/min.

**Figure 4 foods-14-00154-f004:**
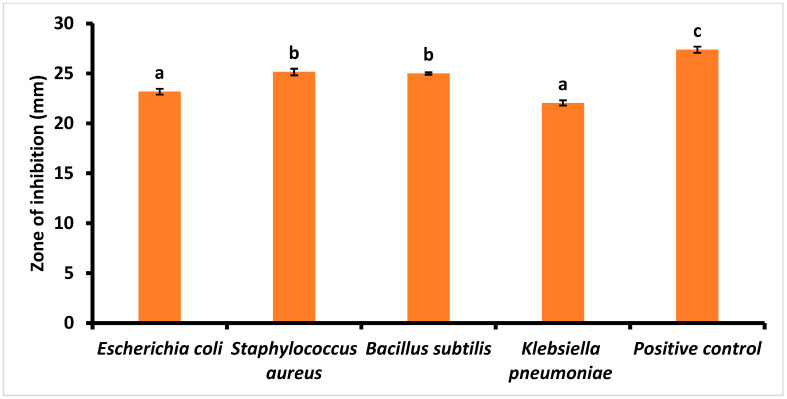
Antibacterial activity of papaya leaf extract against Gram-positive and Gram-negative bacterial species. The letter (a, b, c) above the bars indicate statistically significant differences among the groups (*p* < 0.05).

**Table 1 foods-14-00154-t001:** Nutritional, phytochemical, and antioxidant activity of papaya leaves.

S. No.	Parameter	Concentration
1	Moisture (g/100 g)	15.16 ± 1.15
2	Ash (g/100 g)	3.73 ± 0.21
3	Fat (g/100 g)	4.50 ± 0.86
4	Crude fiber (g/100 g)	9.06 ± 0.92
5	Crude protein (g/100 g)	25.75 ± 0.10
6	Total soluble protein (mg/g)	52.37 ± 0.28
7	Carbohydrates (g/100 g)	41.49 ± 0.10
8	Ca (mg/kg)	1079 ± 0.75
9	Cu (mg/kg)	13.24 ± 0.01
10	Fe (mg/kg)	228.2 ± 0.08
11	K (mg/kg)	4071 ± 0.41
12	Mg (mg/kg)	789.2 ± 0.32
13	Mn (mg/kg)	22.15 ± 0.01
14	Zn (mg/kg)	70.92 ± 0.04
15	Na (mg/kg)	361.2 ± 1.74
16	Total phenolic content (mg GAE/g)	8.85 ± 0.01
17	Total flavonoid content (mg QE/g)	21.00 ± 0.18
18	Tannin content (mg TAE/g)	430 ± 0.08
19	Alkaloid content (g/100 g)	11.4 ± 0.10
20	DPPH (%)	77.55 ± 2.21
21	FRAP (mmol/mg)	25.34 ± 2.45

Mean ± SD of triplicate samples.

**Table 2 foods-14-00154-t002:** Phytochemicals identified in methanolic papaya leaf extract by GC−MS.

S. No	Name	Formula	RT	Area (%)	Biological Effects	References
1	Hydroperoxide, hexyl	C_6_H_14_O_2_	4.01	15.69	Antioxidant	[32]
2	cyclopentasiloxane, decamethyl	C_5_H_10_O	8.28	3.32	Antiviral	[33]
3	Methane, isothiocyanato-	C_2_H_3_NS	12.00	1.41	Antimicrobial	[34]
4	cyclohexasiloxane, decamethyl	C_12_H_36_O_6_Si_6_	12.87	5.51	Antimicrobial	[35]
5	1,1,1,3,5,5,7,7,7-Nonamethyl-3(trimethyl siloxy) tetrasiloxane	C_12_H_36_O_4_Si_5_	15.90	2.90	Antimicrobial	[36]
6	4-Aminosalicylic acid, 3TMS derivative	C_13_H_33_NO_2_S_i3_	18.41	2.69	Antitubercular	[37]
7	1,1,1,5,7,7,7-Heptamethyl-3,3-bis (trimethylsiloxy) tetrasiloxane	C_13_H_39_O_5_S_i6_	20.53	2.05	Antiquorum	[38,39]
8	Piperazine, 1-nitroso	C_4_H_9_N_3_O	22.75	1.21	Antibacterial, Antifungal	[40]
9	Heptane, 4-azido-	C_7_H_15_N_3_	24.81	3.48	Antibacterial	[41]
10	1,1,1,3,5,7,7,7-Octamethyl-3,5-bis (trimethyl siloxy) tetrasiloxane	C_14_H_42_O_5_Si_6_	25.99	1.09	Antimicrobial	[42]
11	9,12,15-Octadecatrienal	C_18_H_30_O	28.97	11.25	Antioxidant	[43]
12	Ethanethioic acid, S-(2-methyl butyl) ester	C_7_H_14_OS	29.38	2.99	Antimicrobial	[44]
13	1,1,1,3,5,5,7,7,7-Nonamethyl-3- (trimethyl siloxy) tetrasiloxane	C_12_H_36_O_4_S_i5_	31.13	1.29	Antimicrobial	[36]
14	1,1,1,3,5,7,7,7-Octamethyl-3,5-bis (trimethyl siloxy) tetrasiloxane	C_14_H_42_O_5_Si_6_	32.99	1.21	Antimicrobial	[42]
15	Hexasiloxane, tetradecamethyl	C_14_H_42_O_5_Si_6_	35.22	1.11	Anticancer	[45]
16	1,1,1,3,5,5,7,7,7-Nonamethyl-3- (trimethyl siloxy) tetra siloxane	C_12_H_36_O_4_S_i5_	38.16	0.89	Antimicrobial	[36]
17	1H-1,2,4-Triazole-3-carboxaldehyde, 5-methyl	C_4_H_5_N_3_O	39.94	1.07	Antifungal	[46]

## Data Availability

The original contributions presented in this study are included in the article. Further inquiries can be directed to the corresponding author.

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
