# Peer review of "Nutritional, Phytochemical, and Antimicrobial Properties of Carica papaya Leaves: Implications for Health Benefits and Food Applications"

_foods, 2025, doi:10.3390/foods14020154_

Round 1
Reviewer 1 Report
Comments and Suggestions for Authors
The article presents the pursuit of valorizing underutilized products in the nutritional and functional field. While the findings are interesting, certain aspects need to be addressed for consideration. These are outlined in the attached file

Author Response
Response to Reviewer X Comments
|
||
1. Summary |
|
|
Thank you very much for taking the time to review this manuscript. Please find the detailed responses below and the corresponding revisions/corrections highlighted/in track changes in the re-submitted files
|
||
2. Questions for General Evaluation |
Reviewer’s Evaluation |
Response and Revisions |
Does the introduction provide sufficient background and include all relevant references? |
Yes/Can be improved/Must be improved/Not applicable |
[Please give your response if necessary. Or you can also give your corresponding response in the point-by-point response letter. The same as below] |
Are all the cited references relevant to the research? |
Yes/Can be improved/Must be improved/Not applicable |
|
Is the research design appropriate? |
Yes/Can be improved/Must be improved/Not applicable |
|
Are the methods adequately described? |
Yes/Can be improved/Must be improved/Not applicable |
|
Are the results clearly presented? |
Yes/Can be improved/Must be improved/Not applicable |
|
Are the conclusions supported by the results? |
Yes/Can be improved/Must be improved/Not applicable |
|
3. Point-by-point response to Comments and Suggestions for Authors
Reviewer 1 |
||
Comments 1: Abstract: The abstract is the first point of contact with the reader after the title. It would be advisable to include more background information regarding the rationale for evaluating papaya leaves. Specifically, where they are consumed or utilized for their bioactive benefits
|
||
Response 1: Thank you for pointing this out. We agree with this comment. Therefore, We revised the background section of the abstract to provide more information about the traditional uses and therapeutic benefits of papaya leaves (PLs), specifically highlighting their consumption for treating inflammation, infections, and various health conditions. We also clarified their potential applications in functional foods. As the maximum word limit for the abstract is 200 words, we have condensed the background accordingly. These changes can be found in the revised manuscript on (page 1, Paragraph: Abstract, and are highlighted in red)
|
||
Comments 2: Introduction Lines 38–41 require proper referencing to support the information presented. There are many claims that are not cited, such as: "PLs have been found to assist in regulating β-cells for insulin release in diabetic patients." Additionally, is this argument relevant to the purpose of the study?
|
||
Response 2: Thank you for your valuable feedback. We have added the appropriate reference to support the information. Regarding the claim 'PLs have been found to assist in regulating β-cells for insulin release in diabetic patients,' we agree that this argument may not be directly related to the primary focus of the study, so we have deleted it. These changes can be found in the revised manuscript on (Page 1 and 2, Paragraph: Introduction, and are highlighted in red).
Comments 3: Lines 58-61, idem: The introduction mentions numerous benefits associated with papaya leaves; however, many of them lack proper substantiation and do not contribute to the presentation of the problem, especially considering that this article does not evaluate biological effects. It is recommended to establish paragraphs that clearly outline the central topic—papaya leaves—the current issue at hand, the rationale behind the study, and the corresponding research question or objective.
Response 3: Thank you for your insightful comment. We acknowledge that the introduction previously mentioned several benefits of papaya leaves, some of which were not adequately substantiated, especially since our article does not focus on evaluating the biological effects of papaya leaves. In response, we have removed these statements from the introduction and revised the section to more clearly outline the central topic—papaya leaves—along with the current issue, the rationale behind our study, and the specific research objective. These changes help ensure that the introduction is more focused and aligned with the aims of the manuscript.(Page 2, Paragraph: Introduction, and are highlighted in red)
Comments 4: Methods In line 127, it should say FOLIN
Response 4: Thank you for your comment. We have corrected the term "Folin" instead of "Foilin." The revised text can be found in the updated manuscript on (Page 3, Paragraph (2.2.3.1): Total phenolic content (TPC), and are highlighted in red)
Comments 5: 2.2.3.1. It was expressed as gallic acid equivalents, but was gallic acid also used to construct the standard curve?
Response 5: Thank you for your comment. Yes, gallic acid was used to construct the standard curve for the total phenolic content (TPC) assay. We have clarified this point in the revised manuscript to ensure accuracy and transparency. The updated information can be found in (Page 3, Paragraph (2.2.3.1): Total phenolic content (TPC), and are highlighted in red)
Comments 6 2.2.3.3. In the determination of tannins, is this method sufficient to accurately measure tannins? How is it ensured that the detected compounds are not chlorogenic acid, gallic acid, or others, but rather polymeric tannins
Response 6: Thank you for your comment. The method employed for tannin determination provides an estimate of the total tannin content, which includes polymeric tannins. However, it does not distinguish between specific tannin subclasses such as gallic acid, chlorogenic acid, or other phenolic compounds. While tannic acid was used as the reference standard for the standard curve, additional techniques, such as HPLC or mass spectrometry, could be used for more specific identification and quantification of polymeric tannins.
Comments 7: Results Although brackets are being used for citations, it is possible to include the author's name when mentioning them. For example, in line 232, you could write: "Joseph et al."…
Response 7: Thank you for your valuable suggestion. We have revised the citations in the Results section to appropriately include the authors' names where relevant. This change has been consistently applied throughout the manuscript to improve readability and adherence to citation guidelines. (Please refer to the revised manuscript)
Comments 8: Lines 233–235: The measurement of soluble protein does not imply that it is papain or its activity. The text should be limited to discussing only what was actually evaluated.
Response 8: Thank you for your valuable suggestion. We have revised the text to focus solely on the evaluated TSP content and its contribution to health benefits, without implying papain activity. The updated text (The TSP content was 52.37 mg/g that highlights the potential of PLs as a source of bioavailable protein. These protein contribute to digestive and therapeutic benefits and are absorbed and utilized directly by body) can be found in the revised manuscript on (page 6, Paragraph (3.1), and highlighted in red).
Comments 9: The expression: "The fatty acids in PLs are associated with wound healing and immunity, making them very useful when added to other fatty acid-rich foods [21]" is unnecessary, as this was not evaluated, and no data on fatty acids are presented in the study.
Response 9: Thank you for your feedback. We acknowledge that the statement regarding fatty acids is not relevant, as no data on fatty acids were evaluated or presented in this study. We have removed this line from the manuscript. (Page 6, Paragraph 3.1).
Comments 10: Table 1 raises concerns: the components of protein, carbohydrates, lipids, ash, and fats should collectively account for 100% of the matrix (or close to it). If they do not, it may indicate inaccuracies in the data or missing components that need to be addressed.
Response 10: Thank you for bringing this to our attention. Upon review, we identified an error in the carbohydrate values presented in Table 1. We have corrected these values and updated Table 1 accordingly in the revised manuscript. The corrected data accurately reflects the composition of papaya leaves. These updates can be found in the revised manuscript on (page 6 Table 1)
Comments 11: Conclusion The phrase "All the obtained results indicate the heterogeneity of PLs" is neither informative nor relevant in the conclusion. It lacks specificity and does not add value to summarizing the findings or implications of the study
Response 11: Thank you for your feedback. We agree that the phrase "All the obtained results indicate the heterogeneity of PLs" lacked specificity and did not contribute meaningfully to the conclusion. We have revised the conclusion (Page 11, Paragraph 4, and are highlighted in red)
Reviewer 2
Comments 1: At what stage of development were the leaves collected
Response 1: Thank you for your comment. Fresh young PLs leaves at the tender stage were collected. This information has been included in the revised manuscript to provide clarity on the stage of leaf development (Page 2, Paragraph 2.1 and are highlighted in red)
Comments 2: At what temperature and for how long were the samples stored exactly? The statement "in a cool place" is too general
Response 2: Thank you for your comment. The prepared powder was stored in an airtight vessel at 4°C for 24 h before subsequent analysis. This clarification has been made in the revised manuscript. (Page 2, Paragraph 2.1 and are highlighted in red)
Comments 3: Ratio of methanol to papaya leaf powder by weight?
Response 3: Thank you for your comment. We have clarified the extraction procedure in the revised manuscript. Specifically, we used a 1:10 weight-to-volume (w/v) ratio. This clarification has been incorporated into the sample extraction (Page 3, Paragraph 2.2.1, and are highlighted in red)
Comments 4: What was the criterion for concentration?
Response 4: Thank you for your comment. The criterion for concentration was to remove excess solvent under reduced pressure using a rotary evaporator at 45°C, ensuring a more concentrated extract for further analysis. These revisions can be found in the updated Methods section of the manuscript (Page 3, Paragraph 2.2.1, and are highlighted in red)
Comments 5: What device was the measurement taken on?
Response 5: Thank you for your comment. We utilized the PerkinElmer Lambda 35 UV/Vis Spectrophotometer for our analyses. We have included this information in the revised manuscript to provide clarity on the instrumentation used in our study. (Please refer to Page 3, Paragraph 2.2.3.1, and are highlighted in red)
Comments 6: What device was the measurement taken on?
Response 6: Thank you for your comment. We utilized the PerkinElmer Lambda 35 UV/Vis Spectrophotometer for our analyses. We have included this information in the revised manuscript to provide clarity on the instrumentation used in our study. (Please refer to Page 3, Paragraph 2.2.3.2, and are highlighted in red)
Comments 7: What device was the measurement taken on?
Response 7: Thank you for your comment. We utilized the PerkinElmer Lambda 35 UV/Vis Spectrophotometer for our analyses. We have included this information in the revised manuscript to provide clarity on the instrumentation used in our study. (Please refer to Page 4, Paragraph 2.2.3.3, and are highlighted in red)
Comments 8: What units was the result given in?
Response 8: Thank you for your comment. The alkaloid content was expressed as a g/100g. We have clarified this in the revised method, specifying that the results are expressed as a g/100g of the dry weight of the sample. (Please refer to Page 4, Paragraph 2.2.3.4, and are highlighted in red)
Comments 9: How was the sample introduced for analysis?
Response 9: Thank you for your feedback. The methanolic PLs extract was subjected to GC-MS analysis, we have included this information in the revised manuscript (Please refer to Page 4, Paragraph 2.2.3.5, and are highlighted in red)
Comments 10: Error with column parameters
Response 10: Thank you for your feedback. We have reviewed and updated column parameters for accuracy. The gas chromatography-mass spectrometry (GC-MS) analysis was conducted using an Elite-1 column PE5 (100% dimethylpolysiloxane) with the following specifications: 30 m ×, 0.25 mm,×0.25 µm (Please refer to Page 4, Paragraph 2.2.3.5, and are highlighted in red)
Comments 11: What device was the measurement taken on?
Response 11: Thank you for your comment. We utilized the PerkinElmer Lambda 35 UV/Vis Spectrophotometer for our analyses. We have included this information in the revised manuscript to provide clarity on the instrumentation used in our study. (Please refer to Page 4 Paragraph 2.2.4.1, and are highlighted in red)
Comments 12: What device was the measurement taken on?
Response 12: Thank you for your comment. We utilized the PerkinElmer Lambda 35 UV/Vis Spectrophotometer for our analyses. We have included this information in the revised manuscript to provide clarity on the instrumentation used in our study. (Please refer to Page 4 Paragraph 2.2.4.2, and are highlighted in red)
Comments 13: Describe in more detail sample preparation for antimicrobial activity analyses.
Response 13: Thank you for your suggestion. The sample preparation for antimicrobial activity analysis was described in the sample extraction method (Paragraph 2.2.1, Page 3), where the process of preparing the sample extract was detailed.
Comments 14: The chapter title should also include a discussion
Response 14: Thank you for your comment. We have revised the manuscript to include a combined "Results and Discussion" section, as per your suggestion. This section can be found on (Page 5, Paragraph 3, and highlighted in red).
Comments 15: Standardize the units, instead of %, give mg/kg-1
Response 15: Thank you for your comment. We have standardized the units in the manuscript to (g/100g). This change has been applied consistently throughout the revised manuscript, specifically in Pages 5 and 6, Paragraph 3.1.
Comments 16: Results based on fresh or dry weight? Standardize units as above
Response 16: Thank you for your comment. We have standardized the units in Table 1 (g/100g) in the revised manuscript, specifically in (Page 6, Table 1, and highlighted in red).
Comments 17: No references to antioxidant activity results
Response 17: Thank you for your comment. We have incorporated the reference to antioxidant activity results in the revised manuscript Please refer to (Page 7 Paragraph 3.1 and highlighted in red) |
Reviewer 2 Report
Comments and Suggestions for Authors
The content of the manuscript "Nutritional, Phytochemical and Antimicrobial Properties of Carica papaya Leaves Implications for Health Benefits and Food Applications" is related to the possibilities of using papaya leaves as a component of functional food due to its high health-promoting properties. The authors presented their research correctly, but some corrections are required to maintain the clarity of the manuscript. The suggested comments are included in the attached file.

Author Response
Response 2: Thank you for your valuable feedback. We have added the appropriate reference to support the information. Regarding the claim 'PLs have been found to assist in regulating β-cells for insulin release in diabetic patients,' we agree that this argument may not be directly related to the primary focus of the study, so we have deleted it. These changes can be found in the revised manuscript on (Page 1 and 2, Paragraph: Introduction, and are highlighted in red).
Comments 3: Lines 58-61, idem: The introduction mentions numerous benefits associated with papaya leaves; however, many of them lack proper substantiation and do not contribute to the presentation of the problem, especially considering that this article does not evaluate biological effects. It is recommended to establish paragraphs that clearly outline the central topic—papaya leaves—the current issue at hand, the rationale behind the study, and the corresponding research question or objective.
Response 3: Thank you for your insightful comment. We acknowledge that the introduction previously mentioned several benefits of papaya leaves, some of which were not adequately substantiated, especially since our article does not focus on evaluating the biological effects of papaya leaves. In response, we have removed these statements from the introduction and revised the section to more clearly outline the central topic—papaya leaves—along with the current issue, the rationale behind our study, and the specific research objective. These changes help ensure that the introduction is more focused and aligned with the aims of the manuscript.(Page 2, Paragraph: Introduction, and are highlighted in red)
Comments 4: Methods In line 127, it should say FOLIN
Response 4: Thank you for your comment. We have corrected the term "Folin" instead of "Foilin." The revised text can be found in the updated manuscript on (Page 3, Paragraph (2.2.3.1): Total phenolic content (TPC), and are highlighted in red)
Comments 5: 2.2.3.1. It was expressed as gallic acid equivalents, but was gallic acid also used to construct the standard curve?
Response 5: Thank you for your comment. Yes, gallic acid was used to construct the standard curve for the total phenolic content (TPC) assay. We have clarified this point in the revised manuscript to ensure accuracy and transparency. The updated information can be found in (Page 3, Paragraph (2.2.3.1): Total phenolic content (TPC), and are highlighted in red)
Comments 6 2.2.3.3. In the determination of tannins, is this method sufficient to accurately measure tannins? How is it ensured that the detected compounds are not chlorogenic acid, gallic acid, or others, but rather polymeric tannins
Response 6: Thank you for your comment. The method employed for tannin determination provides an estimate of the total tannin content, which includes polymeric tannins. However, it does not distinguish between specific tannin subclasses such as gallic acid, chlorogenic acid, or other phenolic compounds. While tannic acid was used as the reference standard for the standard curve, additional techniques, such as HPLC or mass spectrometry, could be used for more specific identification and quantification of polymeric tannins.
Comments 7: Results Although brackets are being used for citations, it is possible to include the author's name when mentioning them. For example, in line 232, you could write: "Joseph et al."…
Response 7: Thank you for your valuable suggestion. We have revised the citations in the Results section to appropriately include the authors' names where relevant. This change has been consistently applied throughout the manuscript to improve readability and adherence to citation guidelines. (Please refer to the revised manuscript)
Comments 8: Lines 233–235: The measurement of soluble protein does not imply that it is papain or its activity. The text should be limited to discussing only what was actually evaluated.
Response 8: Thank you for your valuable suggestion. We have revised the text to focus solely on the evaluated TSP content and its contribution to health benefits, without implying papain activity. The updated text (The TSP content was 52.37 mg/g that highlights the potential of PLs as a source of bioavailable protein. These protein contribute to digestive and therapeutic benefits and are absorbed and utilized directly by body) can be found in the revised manuscript on (page 6, Paragraph (3.1), and highlighted in red).
Comments 9: The expression: "The fatty acids in PLs are associated with wound healing and immunity, making them very useful when added to other fatty acid-rich foods [21]" is unnecessary, as this was not evaluated, and no data on fatty acids are presented in the study.
Response 9: Thank you for your feedback. We acknowledge that the statement regarding fatty acids is not relevant, as no data on fatty acids were evaluated or presented in this study. We have removed this line from the manuscript. (Page 6, Paragraph 3.1).
Comments 10: Table 1 raises concerns: the components of protein, carbohydrates, lipids, ash, and fats should collectively account for 100% of the matrix (or close to it). If they do not, it may indicate inaccuracies in the data or missing components that need to be addressed.
Response 10: Thank you for bringing this to our attention. Upon review, we identified an error in the carbohydrate values presented in Table 1. We have corrected these values and updated Table 1 accordingly in the revised manuscript. The corrected data accurately reflects the composition of papaya leaves. These updates can be found in the revised manuscript on (page 6 Table 1)
Comments 11: Conclusion The phrase "All the obtained results indicate the heterogeneity of PLs" is neither informative nor relevant in the conclusion. It lacks specificity and does not add value to summarizing the findings or implications of the study
Response 11: Thank you for your feedback. We agree that the phrase "All the obtained results indicate the heterogeneity of PLs" lacked specificity and did not contribute meaningfully to the conclusion. We have revised the conclusion (Page 11, Paragraph 4, and are highlighted in red)
Reviewer 2
Comments 1: At what stage of development were the leaves collected
Response 1: Thank you for your comment. Fresh young PLs leaves at the tender stage were collected. This information has been included in the revised manuscript to provide clarity on the stage of leaf development (Page 2, Paragraph 2.1 and are highlighted in red)
Comments 2: At what temperature and for how long were the samples stored exactly? The statement "in a cool place" is too general
Response 2: Thank you for your comment. The prepared powder was stored in an airtight vessel at 4°C for 24 h before subsequent analysis. This clarification has been made in the revised manuscript. (Page 2, Paragraph 2.1 and are highlighted in red)
Comments 3: Ratio of methanol to papaya leaf powder by weight?
Response 3: Thank you for your comment. We have clarified the extraction procedure in the revised manuscript. Specifically, we used a 1:10 weight-to-volume (w/v) ratio. This clarification has been incorporated into the sample extraction (Page 3, Paragraph 2.2.1, and are highlighted in red)
Comments 4: What was the criterion for concentration?
Response 4: Thank you for your comment. The criterion for concentration was to remove excess solvent under reduced pressure using a rotary evaporator at 45°C, ensuring a more concentrated extract for further analysis. These revisions can be found in the updated Methods section of the manuscript (Page 3, Paragraph 2.2.1, and are highlighted in red)
Comments 5: What device was the measurement taken on?
Response 5: Thank you for your comment. We utilized the PerkinElmer Lambda 35 UV/Vis Spectrophotometer for our analyses. We have included this information in the revised manuscript to provide clarity on the instrumentation used in our study. (Please refer to Page 3, Paragraph 2.2.3.1, and are highlighted in red)
Comments 6: What device was the measurement taken on?
Response 6: Thank you for your comment. We utilized the PerkinElmer Lambda 35 UV/Vis Spectrophotometer for our analyses. We have included this information in the revised manuscript to provide clarity on the instrumentation used in our study. (Please refer to Page 3, Paragraph 2.2.3.2, and are highlighted in red)
Comments 7: What device was the measurement taken on?
Response 7: Thank you for your comment. We utilized the PerkinElmer Lambda 35 UV/Vis Spectrophotometer for our analyses. We have included this information in the revised manuscript to provide clarity on the instrumentation used in our study. (Please refer to Page 4, Paragraph 2.2.3.3, and are highlighted in red)
Comments 8: What units was the result given in?
Response 8: Thank you for your comment. The alkaloid content was expressed as a g/100g. We have clarified this in the revised method, specifying that the results are expressed as a g/100g of the dry weight of the sample. (Please refer to Page 4, Paragraph 2.2.3.4, and are highlighted in red)
Comments 9: How was the sample introduced for analysis?
Response 9: Thank you for your feedback. The methanolic PLs extract was subjected to GC-MS analysis, we have included this information in the revised manuscript (Please refer to Page 4, Paragraph 2.2.3.5, and are highlighted in red)
Comments 10: Error with column parameters
Response 10: Thank you for your feedback. We have reviewed and updated column parameters for accuracy. The gas chromatography-mass spectrometry (GC-MS) analysis was conducted using an Elite-1 column PE5 (100% dimethylpolysiloxane) with the following specifications: 30 m ×, 0.25 mm,×0.25 µm (Please refer to Page 4, Paragraph 2.2.3.5, and are highlighted in red)
Comments 11: What device was the measurement taken on?
Response 11: Thank you for your comment. We utilized the PerkinElmer Lambda 35 UV/Vis Spectrophotometer for our analyses. We have included this information in the revised manuscript to provide clarity on the instrumentation used in our study. (Please refer to Page 4 Paragraph 2.2.4.1, and are highlighted in red)
Comments 12: What device was the measurement taken on?
Response 12: Thank you for your comment. We utilized the PerkinElmer Lambda 35 UV/Vis Spectrophotometer for our analyses. We have included this information in the revised manuscript to provide clarity on the instrumentation used in our study. (Please refer to Page 4 Paragraph 2.2.4.2, and are highlighted in red)
Comments 13: Describe in more detail sample preparation for antimicrobial activity analyses.
Response 13: Thank you for your suggestion. The sample preparation for antimicrobial activity analysis was described in the sample extraction method (Paragraph 2.2.1, Page 3), where the process of preparing the sample extract was detailed.
Comments 14: The chapter title should also include a discussion
Response 14: Thank you for your comment. We have revised the manuscript to include a combined "Results and Discussion" section, as per your suggestion. This section can be found on (Page 5, Paragraph 3, and highlighted in red).
Comments 15: Standardize the units, instead of %, give mg/kg-1
Response 15: Thank you for your comment. We have standardized the units in the manuscript to (g/100g). This change has been applied consistently throughout the revised manuscript, specifically in Pages 5 and 6, Paragraph 3.1.
Comments 16: Results based on fresh or dry weight? Standardize units as above
Response 16: Thank you for your comment. We have standardized the units in Table 1 (g/100g) in the revised manuscript, specifically in (Page 6, Table 1, and highlighted in red).
Comments 17: No references to antioxidant activity results
Response 17: Thank you for your comment. We have incorporated the reference to antioxidant activity results in the revised manuscript Please refer to (Page 7 Paragraph 3.1 and highlighted in red)
|
4. Response to Comments on the Quality of English Language |
Point 1: |
Response 1: (in red) |
5. Additional clarifications |
[Here, mention any other clarifications you would like to provide to the journal editor/reviewer.] |